# The Sperm Small RNA Transcriptome: Implications beyond Reproductive Disorder

**DOI:** 10.3390/ijms232415716

**Published:** 2022-12-11

**Authors:** Sze Yan Chan, Crystal Wing Tung Wan, Tin Yu Samuel Law, David Yiu Leung Chan, Ellis Kin Lam Fok

**Affiliations:** 1School of Biomedical Sciences, Faculty of Medicine, The Chinese University of Hong Kong, Hong Kong SAR, China; 2Department of Obstetrics and Gynecology, Faculty of Medicine, The Chinese University of Hong Kong, Hong Kong SAR, China; 3The Chinese University of Hong Kong Joint Laboratory for Reproductive Medicine, West China Second University Hospital, Sichuan University, Chengdu 610017, China

**Keywords:** male fertility, paternal epigenetic inheritance, sperm epigenome, ICSI

## Abstract

Apart from the paternal half of the genetic material, the male gamete carries assorted epigenetic marks for optimal fertilization and the developmental trajectory for the early embryo. Recent works showed dynamic changes in small noncoding RNA (sncRNA) in spermatozoa as they transit through the testicular environment to the epididymal segments. Studies demonstrated the changes to be mediated by epididymosomes during the transit through the adluminal duct in the epididymis, and the changes in sperm sncRNA content stemmed from environmental insults significantly altering the early embryo development and predisposing the offspring to metabolic disorders. Here, we review the current knowledge on the establishment of the sperm sncRNA transcriptome and their role in male-factor infertility, evidence of altered offspring health in response to the paternal life experiences through sperm sncRNA species and, finally, their implications in assisted reproductive technology in terms of epigenetic inheritance.

## 1. Introduction

The sperm inherit the paternal half of the genetic material along with epigenetic instructions for the embryo development and inheritable traits. These epigenetic instructions include DNA methylation, histone/protamine marks and a plethora of RNA species. The production of male gametes is a tightly orchestrated cellular process, and the sperm epigenome and transcriptome are established at different phases of sperm production and maturation along the male reproductive tract. These events safeguard the integrity of the paternal genetic and epigenetic information that is delivered to the oocyte during fertilization. In the past decade, emerging studies revealed the importance of small RNA species (<200 nt in length) enriched in the sperm and their roles in paternal epigenetic inheritance. This review will first describe the current knowledge on the establishment of the sperm epigenome and the sncRNAs carried by the sperm, herein referred to as the sperm sncRNA transcriptome. We will then focus on how the sperm sncRNA transcriptome changes according to environmental insults and their effects on fertility and the offspring. Lastly, we will discuss the implications of the sperm sncRNA transcriptome in assisted reproductive technologies (ART).

## 2. Establishment of the Sperm Epigenome

The first phase of sperm production is called spermatogenesis. In both rodents and humans, it starts with a mitotic division of diploid spermatogonial stem cells at the basal lamina of the seminiferous tubule in the testis [1]. The pair of new daughter cells either remain as part of the stem cell pool or migrate away from the basal compartment as chained pairs to follow the differentiation path [2]. The differentiating spermatogonia then enter the meiotic cycle and become the primary spermatocytes. After two rounds of meiotic division I/II without DNA replication, the germ cells are now haploid, denouncing the end of the meiotic cycle and becoming round spermatids, and then proceed to post-meiotic differentiation known as spermiogenesis [3].

While mitotic and meiotic germ cells undergo sophisticated networks of cellular processes that involve the reprogramming of DNA methylation, the histone marks and sncRNA, the epigenome and transcriptome of testicular spermatozoa are mainly established during the spermiogenesis that takes place at the adluminal compartment of the seminiferous tubules. During this process, a wave of transcriptional activity is observed in the round spermatids [4] during its extensive metamorphosis to transform into elongated spermatids [5]. It is believed that the transcriptional activity plays a pivotal role in establishing the repertoire of RNA species in the testicular spermatozoa expressed by the germ cell per se. The paternal genome is then reorganized through the process of histone-to-protamine transition, followed by nuclear condensation, which gives rise to the unique and compact nucleus seen in the spermatozoa. In parallel to the histone-to-protamine transition, the testicular spermatozoa further become transcriptionally inactive, and the bulk of their cytoplasmic content, containing typical cellular organelles including ribosomes, is shed as the residual body [6]. Within the condensing nucleus, the hyperacetylation and phosphorylation of histones result in the relaxation of DNA strands around the nucleosomes and thereby the chromatin structure as well. Chromatin remodeling occurs as nuclear histones are temporarily replaced with transition proteins, followed by subsequent protamine transition to further condense the paternal genetic material into a unique chromatin structure significantly smaller than the somatic cells at the interface [7]. The aberrant disruption of histone-patterning in sperm causes not only fertility problems but also defects in the development of the offspring [8,9]. Depending on the species, there is an estimate of 1% of histone in the sperm chromatin left in mice that is not replaced with protamine, and the estimate is roughly 15% in humans [10]. Evidence from chromatin immunoprecipitation (ChIP)-sequencing showed that the remaining histones are retained in the promoter region of genes linked to fertility and embryo development [11,12]. This is likely to facilitate embryo development in a timely manner, as studies found that, unlike paternal protamines, the paternal histones do not get replaced with oocyte-derived histones upon fertilization [13,14,15].

As protamine-bound genes remain inaccessible, the histone-code generated at the end of spermiogenesis in turn serves as an epigenetic instruction that can impact the offspring health (intergenerational) as well as subsequent generations (transgenerational) [16,17]. Up until recently, it was believed that the sperm chromatin compaction was fully established once it exited the testicular environment, as genome-wide methylation remained stable between the testicular spermatozoa and the cauda epididymal sperm [18]. New studies revealed continuous differential histone retention sites between the caput sperm and cauda sperm of the epididymis [19], suggesting that the alteration of histone marks acquired during the epididymis transit may play a role in the early embryo development and predisposition to offspring health.

Aside from histone marks, sperm also carries a distinct epigenetic profile made up of RNA species. The general consensus had viewed the sperm that has a condensed nucleus and sheds its cytoplasmic residue as transcriptionally inert. The insufficient ribosomal RNA species in sperm also suggested that they are devoid of translational activity [20]. Over the years, it was assumed that the sperm RNAs were mere remnants of spermatogenesis that were not fully discarded along with the cytoplasmic residue. Even so, after the first RNA was identified in the sperm of humans, mice and rats [21], numerous reports further identified RNA species in the male gamete, both coding [22,23,24] and noncoding [25,26]. Sperm-RNA transcripts that are normally not present in the oocyte can also be identified upon fertilization [27], suggesting contributions of the sperm RNA to the embryo, and subsequent studies have also reported the pivotal roles of sperm-RNA in embryonic development [23,28,29,30,31,32,33].

The small RNA (sRNA) profiling of mice revealed an extensive makeover of sRNA species between testicular sperm, caput and cauda sperm [34], where Piwi-interacting RNAs (piRNAs) account for about 85% of sRNA species in the testicular spermatozoa, followed by 10% of microRNA (miRNA). Upon entering the epididymis, transfer RNA-derived small RNAs (tsRNAs) become the dominant sRNA, with 75% of total sRNA species in the caput sperm and more than 85% in the cauda sperm. The level of miRNAs remains fairly constant during the epididymal transition, which represents 5% of the total sRNA, making it the second most abundant sRNA species in cauda sperm. The profiling of miRNAs from caput, corpus and caudal sperm reported a loss of 113 miRNAs, as well as a gain of 115 miRNAs, including a significant increase in miRNAs natively expressed in immature sperm [35].

Sperm are transcriptionally inactive after exiting the testis, despite RNA polymerase having been found in epididymal sperm in both bulls and mice [36,37]. Currently, there is no evidence of endogenous activity of sRNA transcription occurrence in the testicular spermatozoa to cauda sperm, which could explain the dynamic change in the sperm sRNA profile, nor is there evidence of the endogenous cleaving of intact transfer RNA (tRNA) in testicular spermatozoa, which could explain the abundance of tsRNA starting in the caput sperm. This raises suspicions that the abundance of tsRNA could be trafficked to the maturing sperm through extrinsic factors during their transit in the epididymis.

Early studies using the electron microscopy of sperm from Chinese hamsters showed small membranous vesicles over the acrosome of caudal sperm [38]. High-resolution microscopy, in turn, showed an interaction between extracellular vesicles (EVs) and the cytoplasmic droplets on the caudal sperm [39]. These EVs, later referred to as epididymosomes, are characterized, and their interaction with epididymal sperm had been established by Sullivan et al. through co-incubation and surface-protein labeling [40]. The binding of epididymosomes results in the incorporation of several proteins found in epididymosomes to the sperm membrane. This could either be because the epididymosomes do not fuse completely with the sperm to allow for the transfer to the whole protein repertoire or because only a subpopulation of EVs are delivered to the sperm, as the epididymosomes is a heterogeneous pool of vesicles.

In turn, using in vivo metabolic labeling of RNAs, Sharma et al. showed that RNAs that specifically synthesized the caput epididymis were taken up by the caudal sperm [34]. The co-incubation of testicular spermatozoa with caput epididymosomes also results in a significant increase in the abundance of tsRNAs and miRNAs. tRNA fragments (tRFs) corresponding to the 5′ end of tRNA Glu-CTC and Val-CTC, in particular, were increased 4–10 fold in testicular sperm co-incubated with caput epididymosomes when compared to the mock-treated control. This confirmed the ability of sperm to receive not only the protein but also the RNA cargo of the epididymosomes, thereby modifying both the sperm proteomic and RNA profiles.

## 3. Sperm sncRNA Transcriptome and Fecundity/Fertility

The sperm sncRNA transcriptome has been implicated in fecundity/fertility via their involvement in sperm production and sperm functions. On one hand, these sncRNAs take part in spermatogenesis and remain in the sperm. In this case, the sncRNAs in sperm play a passive role in fecundity/fertility by acting as a biomarker of spermatogenic defects. On the other hand, accumulating evidence (reviewed below) suggests an association between sperm sncRNAs and sperm functions, including the contributions to embryo development. These results suggest an active role of the sperm sncRNA transcriptome on fecundity/fertility.

Among the sncRNAs, miRNA is the most extensively studied sncRNA species owing to its essential roles in regulating gene expressions at the post-transcriptional level. In men, reduced expressions of sperm miR-34b/c and miR-449a are often reported in subfertile individuals [41,42,43] and infertile patients associated with spermatogenic failure [44]. The miR-34b/c and miR-499a/b/c are exclusively expressed in testicular germ cells [45,46]. They share the same seed sequence and common mRNA targets involved in, e.g., the Transcription Factor-Retinoblastoma Protein (E2F-pRb) Pathway [45], which plays a major role in the regulation of the mammalian cell cycle [47]. The regulation of this pathway by miR-34 and miR-449 allows for the spermatogonium to exit the mitotic phase and enter the meiotic stage of spermatogenesis [48]. However, paternal miR-34b/c and miR-449 are dispensable for fertilization and embryo development as the injection of round spermatids harboring double knockouts of said miRNAs to wild-type oocytes give rise to embryos that are fully capable of developing to term with normal liveborn in a mouse study [49]. Therefore, the differential expression of these miRNAs in the sperm appears to reflect the spermatogenic defects during the establishment of the sncRNA transcriptome rather than playing functional roles in sperm functions, fertilization and embryo development.

Unlike miRNA, and despite being the most abundant sRNA species in spermatogenic cells [34,50,51], the function of piRNA is more specific in the silencing of transposable elements [52,53], which plays an essential role in protecting and stabilizing the genome and reproductive fitness of the male gamete during spermatogenesis [54,55]. Recent studies have shown that the deletion of piRNA clusters of the mouse genome results in sperm defects such as capacitation, zona pellucida penetration and motility. Sperm with the chromosome 18 piRNA cluster deleted presents with acrosome dysgenesis, severe sperm head dysmorphology and failure in fertilization through insemination and in vitro fertilization (IVF) due to impaired motility [56]. Another study showed that the deletion of chromosome 6 piRNA clusters resulted in defects in acrosome reaction and sperm motility [57]. The microinjection of the sperm head to the normal oocyte showed that mature sperm with such deletion retained their fertilizing capability, albeit with a delayed first cleavage and eventual embryonic lethality at the two-cell stage [57], suggesting that chromosome 6 piRNAs are essential not only for normal spermatogenesis but also for early embryonic development.

In men, microarray analysis revealed more than one hundred differentially expressed piRNAs in the sperm samples of asthenozoospermic patients compared to the normozoospermia cohort [58]. Among the significantly upregulated piRNAs, piR-26951 was predicted to target transcripts encoding flagella-associated protein 45 (CFAP45) and bromodomain-containing protein 2 (BRD2), both found to be differentially expressed in the sperm of patients with asthenospermia [59,60]. In another study, the general abundance and types of piRNAs were found to be reduced in the sperm of asthenozoospermia compared to normozoospermia. An analysis revealed 17 piRNAs with a more-than-twofold decrease, to which piR-1207 and piR-2107 were significantly downregulated in asthenozoospermia patients. A reduced expression of the PIWI protein MitoPLD, which is responsible for piRNA biogenesis, was also observed in the sperm samples of asthenozoospermia, potentially explaining the reduced abundance of piRNA in the infertile group [61]. A study of sperm from oligoasthenozoospermia patients reported that reduced expressions of piR-31704 and piR-39888 correlated with a lower sperm motility, a lower rate of good-quality embryo and a significantly reduced sperm concentration and fertilization rate after ICSI [62].

The downregulation of piRNAs in mature sperm is therefore associated with sperm defects, and it has yet to be explored whether the reduced fertilization and embryo quality are caused by the piRNAs carried by the sperm or by the sperm defect stemming from altered piRNA expression during spermatogenesis.

tRFs are sncRNAs derived from tRNAs through enzymatic processes. These fragments range from 14 to 40 nt-long, depending on their site of cleavage by different enzymes including Dicer, Angiogenin and members of the Ribonuclease A family. Cleaving at the anticodon loop by Dicer produces 5′- and 3′-tRNA halves that range from ~30 to 40 nt, whereas shorter tRFs of 14–30 nt are produced by cleaving at different sites of the D-loop (5′-tRFs) or the T-loop (3′-tRFs) of the mature tRNA. Additional fragment subtypes include tRF-1, tRF-2 and i-tRFs, which are generated from the 3′-end of the precursor tRNA, the anticodon loop and the anticodon loop straddled with parts of the D- and T-loop [63]. Due to the current lack of standardized nomenclature for tRFs and the variation in how they are referred to in studies, herein, we will refer to tRFs in general as tRNA-derived small RNA (tsRNA), unless specified.

Increasing studies had demonstrated diverse biological functions for tsRNAs in post-transcriptional gene regulation, where they act similarly to miRNA and piRNA by associating with Argonaut and Piwi proteins [64] and modulating cellular behavior [65] and dysregulation in health and disease [66].

The deep sequencing of the sRNA population (<40 nt) in the mature sperm of mice from the cauda epididymis revealed an abundance of sRNAs at 30–34 nt mapped and matched to the tRNA locus and specifically to the 5′ end of tRNA halves [67]. To date, very few studies have examined the involvement of tsRNA in sperm functions and fertility. In an animal study, Sharma et al. identified abundant sRNAs of 28–34 nt from the cauda sperm, predominantly derived from the 5′-ends of tRNAs [68]. Here, the sperm 5′-tRF-Gly-GCC was reported to mediate endogenous retroelements and gene regulation in early embryonic development. A study on porcine spermatogenic cells and ejaculated sperm revealed the presence of tsRNA of 25–35 nt predominantly mapped to the 5′-ends of tRNAs as well [69], and the enrichment of 32–34 nt tsRNA Gln-TTG in the ejaculated sperm is speculated to be involved in early embryonic cleavage by mediating genes related to the cell cycle, nuclear division, organelle organization as well as retrotransposable elements through annotation analysis. The follow-up study utilizing the microinjection of synthetic sequences of tsRNA Gln-TTG to human tripronuclear embryos confirmed their association with early embryo development by regulating developmental and signal-transduction genes, as well as genes related to ncRNA processing [70]. In men, the Gly-GCC, Thr-TGT and Glu-TTC of different tRF subtypes are downregulated in sperm samples undergoing IVF treatment, whereas the upregulation of Pro-AGG, Pro-TGG, Asn-ATT and Arg-GGC is associated with a low rate of good-quality embryos [71]. In summary, current evidence suggests the involvement of tsRNAs in fertility/fecundity via their effect on early embryo development.

## 4. Alteration of the Sperm sncRNA Transcriptome in Response to Environmental Insults

As studies have reported the involvement of the sperm sncRNA transcriptome in sperm functions, reports on the inheritance of sperm sncRNAs to the oocyte continue to expand. Importantly, these inherited sperm sncRNAs play pivotal roles in embryo development and mediate the paternal inheritance that impacts the health of the offspring. It has been hypothesized that these sperm sncRNAs represent an RNA code for the inheritable traits [51], and the effects can be intergenerational or transgenerational [72]. Interestingly, the majority of the findings have suggested that the inheritable traits were attributed to two major RNA species, miRNA and tsRNA. However, the underlying mechanisms behind such inheritance are not fully understood. Moreover, the study of epigenetic inheritance in humans is not always feasible, as it is impossible to directly use the connection due to many other factors, such as nutrition, behavior, physical activity, social interaction and work habits, contributing to changes in biological function and inheritable phenotypes. In turn, animal models such as rodents allow for a greater level of control in terms of the lifetime environment, down to their genotypes [73,74,75]. Here, we review the recent findings on inheritable traits through sperm sRNA species in rodent studies and the correlation with the human cohort.

### 4.1. Alteration of Sperm sncRNA in Chronic Stress

Rodent stress models are established between 2 and 7 weeks across different stages in life, such as early life (postnatal) [76], puberty (4 weeks old) [77,78,79] and adulthood (8 weeks old) [77,80,81], with variable inheritable phenotypes reported owing to diverse protocols and assessments. In a mouse model of early life stress, pups at postnatal day 1 were exposed to unpredictable maternal separation and unpredictable maternal stress (MSUS) over the course of 2 weeks [76], which led to three generations of behavioral despair and depression-like phenotypes, including reduced avoidance, fear and aversion, coupled with insulin hypersensitivity for F1 offspring and hypermetabolism plus reduced body weight for F2 and F3 progenies.

A chronic stress model utilizing random stress factors over the course of 6 weeks during puberty (4 weeks old) programmed the direct descendants with a reduced HPA axis and blunted corticosterone responsivity to acute stress [77]. Alternatively, 7 weeks of social instability by cage shuffling showed a remarkable gender bias of inheritable traits, where only female descendants across three generations harbor elevated anxiety and defective social behavior, independent of the serum corticosterone level, which was elevated exclusively in the F1 females [78]. This gender discrepancy was also observed in the chronic stress model through the sustained elevation of glucocorticoids in adult mice (8 weeks old) [80]. In this model, mice were given water supplemented with corticosterone over the course of 4 weeks. The resultant progenies showed hyper anxiety-like behavior in F1 males, the depression-like phenotype in F2 males and lower levels of anxiety in F2 females. Finally, a depression model established by chronic mild stress factors over the course of 5 weeks in adult mice programmed vulnerability and susceptibility to depressive phenotypes to direct progenies when exposed to stressful situations [81].

Despite the diversified components in establishing these stress models, 16 miRNAs (let-7f/g/i, miR-9, miR-21a, miR-26b, miR-30a, miR-30c, miR-32, miR-103, miR-152, miR-194, miR-200c, miR-204, miR-375, miR-449) in the sperm sncRNA transcriptome were found to overlap in these studies in terms of their altered expressions [76,77,78,79,80,81]. For instance, in the study of Rodgers et al., where they examined the sperm miRNA content between puberty and adult mice, the expressions of nine miRNAs (miR-29c, miR-30a, miR-30c, miR-32, miR-193, miR-204, miR-375, miR-532, miR-698) were significantly increased in both groups in response to chronic stress [77]. The prediction analysis of these miRNAs pointed towards DNA methylation by targeting DNA methyltransferase 3a (DNMT3a) and two other proteins involved in miRNA processing: the trinucleotide repeat containing 6b (Tnrc6b) and metadherin (Mtdh). A follow-up study with the microinjection of the nine miRNAs to wild-type zygote reduced the maternal mRNAs in the early zygote, such as Sirtuin 1 (Sirt1) and Ubiquitin-protein ligase E3A (Ube3a), which were both reported to be involved in chromatin remodeling and neurodevelopmental disorders. The resultant offspring derived from microinjection showed altered gene expressions in relation to the extracellular matrix and collagen in the hypothalamus and recapitulated the dysregulated HPA responsivity, as seen in their earlier study [77,82].

Among the nine miRNA candidates reported in the above-mentioned study [77], the elevated expressions of miR-30a, miR-30c, miR-204 and miR-375 were reported in the sperm sRNA profiles of both early life [76] and adulthood stress models [81]. The microinjection of sperm total RNA allowed for the recapitulation of the altered phenotypes in both early life and adulthood studies. Furthermore, the microinjection of 16 synthetic miRNAs that were upregulated in the F0 sperm of chronic mild stress sires recapitulated the behavioral phenotypes [81]. Gene ontology analysis revealed 78 out of the 264 differentially expressed genes in the embryo to be the direct targets of the 16 injected miRNA candidates involved in neuronal functions such as synaptic plasticity and dendritic spine formation. Finally, the microinjection of miRNA antisense to depression-zygote abolished the inheritable traits, demonstrating the transmission of paternal traumatic experiences through sperm miRNA, altering embryonic development and subsequent offspring gene expressions. Curiously, sperm miRNA profiles were not altered in the F1 offspring from stressed sires established through corticosterone-supplemented feeding [81], corroborating the ceased phenotype in the F2 offspring. Conversely, the altered sperm miRNA profiles persisted in the F1 offspring but not the F2 generation in the postnatal and puberty models [76,79]. Yet, the dysregulated phenotypes were preserved in the F3 offspring as well. Setting aside other mechanisms that permit paternal epigenetic inheritance such as DNA methylation and histone post-translational modifications, the phenotypes established by a certain miRNA candidate or a distinct population of miRNAs and their subsequent target pathways during embryo development may determine the intergenerational and transgenerational inheritance of paternal life experiences.

In men, among the nine miRNA candidates reported by Rodgers et al. [77], miR-29c and miR-30c expressions were reportedly changed in the blood of human male individuals exposed to arithmetic social stress tasks [83], and the increased expression of miR-30c in blood is also associated with a stressful environment such as social defeat and bullying in both mice and men [84], suggesting that the two miRNAs play a role in response to stress, and their elevation in the sperm may play a functional role in predisposing the offspring behavior by regulating gene expressions in the early embryo development. In another study aiming to identify sperm biomarkers to predict the offspring developmental risk and resilience factor, Morgan et al. performed repeated sperm sampling from humans over the course of 6 months to evaluate the dynamic changes in sperm sncRNA, followed by monthly psychological evaluations regarding perceived stress. In their study, they identified five miRNAs (let-7f, miR-181a, miR-4454, miR-6765 and miR-12136) that are associated with perceived stress [85]. Among these, the elevated expression of let-7f overlapped with previous mouse model studies [76,80]. Similarly, in search for common sperm miRNA changes in response to stress, Dickon et al. performed a microarray analysis of sperm sRNAs between men exposed to early life stress (average age of 32.4) and mice (4 weeks old) exposed to social instability by random shuffling between cages for 7 weeks. Here, they found an inverse correlation between the levels of the miR-449 and miR-34 families and the stress scores in the human males. Likewise, the same miRNA family was significantly reduced in the sperm of their mouse model, which persisted to the following F1 generation as well [79].

Together, these studies validated the potent functionality of sperm miRNA upon fertilization, which then triggers a cascade of molecular events that alter the stress reactivity in the next generations of offspring. Finally, contrasting the chronic stress models, male mice receiving a single strong footshock were mated 6 weeks later, and the resultant generation showed no anxiety or depressive-like behaviors. Female offspring were found to have a reduced body weight compared to those of the control sires [86]. No molecular investigation was performed; this showed that a single event exposure adequately brought alteration to the offspring phenotype. Interestingly, as it takes 42 days (6 weeks) to complete one round of spermatogenesis in mice [87], it would have been interesting to investigate the duration of a single event of an environmental insult that could persist over time.

### 4.2. Alteration of Sperm sncRNA in Obesity

With the rise in popularity of western diets owing to inexpensive, calorie-dense food, coupled with technological advances that significantly reduced the need for physical activity, the spread of obesity has come to be labeled as a pandemic. The resultant lifestyle severely impacted fertility in both males [88] and females [89]. In rodent studies, obese models are established by a high-fat diet (HFD) over the course of 9–26 weeks. In the mouse study of Fullston et al., an HFD of 10 weeks resulted in a significant alteration to the transcriptome and miRNA expression in the testicular environment [90]. Among the differentially expressed miRNA in the HFD testis, the expression of miR-133b-3p, miR-196a-5p, miR-205-5p and miR-340-5p was also significantly altered in the matured sperm of the vas deferens. A pathway analysis revealed the four differentially expressed miRNAs to be involved in spermatogenesis, embryo development, insulin signaling and metabolic disorders. Indeed, studies had shown a significant reduction in sperm motility [91], increased sperm DNA damage [92] and impaired embryo development [93,94] as a result of paternal HFD. miR-30a was among the identified candidates that are significantly upregulated in the sperm of cauda and the vas deferens of HFD mice [91] and reported to regulate adipogenesis in both mice and men [95,96], which could be associated with the elevated body weight in the offspring.

Apart from an increased body weight, both impaired glucose tolerance and insulin sensitivity are commonly reported in the F1 [97,98] and F2 offspring of HFD sires [90,99]. These phenotypes were recapitulated in F1 offspring derived using epididymal sperm RNA microinjected to normal zygote [99]. In a study of a 17-week-long western-like diet (high fat and high sugar), miR-19b-3p was significantly upregulated in the epididymal sperm. The resultant F1 offspring from microinjecting synthetic miR-19b-3p to normal zygote partially recapitulated the same phenotypes. Furthermore, when crossing the synthetic miR-19b-3p microinjection-derived F1 males with normal females, the resultant F2 shared a similar partial recapitulation of the phenotypes. Interestingly, the altered expression of sperm miRNA in F0 HFD mice was not recapitulated in the F1 offspring [91], suggesting the intergenerational inheritance of HFD-induced glucose intolerance and insulin sensitivity through sperm miRNA, while the mechanism of the transgenerational inheritance of such phenotypes has yet to be explored.

In another study of a 26-week-long HFD mouse model, Chen et al. [97] revealed significant changes in cauda sperm miRNA (2.28%) and tsRNA (11.53%). In their study, the tsRNAs were mainly 5′ tRNA halves that were 30–34 nt long and mapped to gene promoter regions involved in cellular and molecular events in the early embryo development. The F1 offspring derived from the microinjection of HFD caudal sperm RNA ranging from 30 to 40 nt to normal zygote mimicked the glucose intolerance in F1 offspring derived from whole-sperm head injection, but not the insulin sensitivity. A similar result was observed after the microinjection of HFD sperm total RNA. In contrast to earlier studies, the microinjection of sperm miRNA (15–20 nt) led to embryo lethality, while the microinjection of >40 nt sperm RNA did not result in any phenotype, suggesting that the intergenerational inheritance of glucose metabolism may be mediated by the tsRNAs of the matured sperm. Furthermore, this inheritance of HFD-induced glucose metabolism requires RNA modification on sperm tsRNAs as reported by Zhang et al. [98], where the deletion of tRNA methyltransferase (Dmnt2) abolished the m^5^C and m^2^G modifications in their HFD sperm tsRNAs, as well as the metabolic disorders in the subsequent offspring of Dmnt2^−/−^ HFD sires. This phenomenon was also observed in the HFD model of Chen et al. [97], with a significant increase in the RNA-methylation of m^5^C and m^2^G in sperm tsRNAs compared to a normal diet, and the microinjection of unmodified synthetic 5′ tRF candidates that were most abundant in the HFD sperm did not induce metabolic disorder in the offspring.

In men, sperm profiling between lean and obese individuals showed a marked difference in sncRNA content, along with DNA methylation patterns [100]. The differentially expressed sncRNA included tsRNA, miRNA and piRNA. Among these, sperm miR-133b was previously reported to be significantly upregulated in HFD mice models [90,91]. The miR-133b targets insulin growth factor receptor 1 (Igf-1R), which regulates cell growth and survival. While it is dispensable in germ cells and spermatogenesis, it is associated with Sertoli cell proliferation and adult sperm production [101]. The Igf-1R had also been reported to be essential to preimplantation embryo survival during murine blastocyst formation [102]. The inhibition of Igf-1R by upregulated miR-133b may therefore be associated with infertility. In contrast to mice studies, miR-133b expression was downregulated in obese men when compared to the lean cohort [100]. Instead, sperm miR-195 was significantly upregulated in the obese individuals, and this miRNA had been reported to target Sirt1 [103], which is involved in the methylation and acetylation of the histone code in the porcine zygote. The activation of Sirt1 was also reported to be associated with improved embryo development [104]. The significant upregulation of miR-195 in the obese sperm may be associated with the impairment of implantation and embryo development shown in the mouse HFD model [93].

The accumulated studies on obesity using mouse models reported a diverse set of mature sperm miRNA candidates implicated in inducing metabolic disorder in the offspring. Surprisingly, only a few miRNA candidates overlapped between the studies, and some showed a contradictory expression profile as well, possibly owing to the greater length of time each study employed in establishing the obese models (10 w, 17 w, 20 w, 26 w); the change in the sperm sncRNA profile concomitant with aging [105] might have shrouded the key players in response to the excessive change in diet. Interestingly, miR-19b alone was found to play a larger role in shaping the offspring’s health, as the microinjection of synthetic miR-19b to zygote could partially recapitulate the western-like diet phenotypes in the offspring [99]. On the other hand, the study of Chen et al. suggested that tsRNAs (30–40 nts), but not miRNAs (20–30 nts), are responsible for the inheritable metabolic phenotypes [97]. Currently, the study of sperm tsRNAs is still in its infancy, and the role of sperm sncRNA species in predisposing HFD-induced traits to offspring still needs to be investigated.

### 4.3. Other Inheritable Traits Involving Sperm sncRNAs

In a study of inflammation-induced heritable phenotypes, Zhang et al. showed that offspring derived from inflammatory model sires exhibited obesity and metabolic syndrome-like phenotypes such as an increased body weight, perigonadal fat mass and fat-to-muscle ratio, coupled with a decreased exercise capacity and impaired glucose tolerance [106]. These phenotypes were partially recapitulated in the offspring derived from the intracytoplasmic injection of control cauda sperm along with synthetic tsRNAs, which were most abundantly increased in the sperm of the inflammation-model sires. Contrasting the HFD studies by Chen at al. [97] and Zhang et al. [98], where methylation on sperm tsRNAs was required to induce observable traits in the offspring, the inflammation-induced phenotypes in the offspring could be mediated by unmodified tsRNAs to a moderate degree. Finally, these phenotypes were abolished in mutant mice with the deletion of Angiogenin, which is abundantly expressed in the caput epididymis, suggesting that inflammation-induced inheritable phenotypes in mice were regulated by the tsRNA processed during the caput epididymal transit. Similarly, a reduced expression of tRNA methyltransferase Nsun2 in the epididymis was observed together with an altered sperm tsRNA profile in mice exposed to ethanol vapor. In the chronic ethanol exposure model by Rompala et al. [107], 15 tsRNAs, 8 miRNAs and 5 mitochondrial small RNA expressions were significantly altered in the cauda sperm. The altered sncRNA were predicted to be involved in a wide range of biological functions and scored the highest for transcriptional factors and phosphoproteins. Among the resultant F1 generation, only males showed a reduced ethanol-drinking preference, an increased ethanol sensitivity to the anxiolytic effect and an increased expression of the brain-derived neurotrophic factor (BDNF) in the ventral tegmental area [108,109,110]. In another preconception paternal alcohol use study by Bedi et al. [111], mice exposed to the voluntary consumption of ethanol showed a slight alteration in sperm sncRNA abundance, with a 30% increased enrichment in miRNAs in the alcohol group and a significant upregulation of miR-21 and miR-30, while miR-142 was downregulated. Here, the reduced fetal weight is associated with a decreased placental efficiency in both male and female offspring.

## 5. Origin of Altered Sperm sncRNA Payload

The sperm sncRNA transcriptome undergoes a shift during the epididymal maturation, with a marked increase in tsRNAs but a decrease in piRNAs, while the miRNAs remained constant. Notably, while a plethora of studies have demonstrated the effect of the sperm sncRNA transcriptome on the paternal inheritance, very few have examined when the sperm sncRNA payload was altered. Emerging evidence has suggested an important role of the epididymosomes in conveying the altered sncRNAs into sperm in response to the environmental insults.

In the study by Chan et al. [112], the co-incubation of normal cauda sperm with EVs secreted by corticosterone-treated caput epididymal epithelial cells and subsequent intracytoplasmic injection produced offspring with altered neurodevelopment and recapitulated the physiological phenotypes of progenies from natural mating with stressed sires. The sequencing data of cauda epididymosomes isolated from a postnatal stress mouse model showed a differentially altered expression of small RNAs, of which 70% corresponded to miRNA, 15% corresponded to tsRNA and 15% corresponded to piRNA and snoRNA [113]. Although not reported in other stress model studies, the differentially upregulated miR-31-5p reported in the cauda epididymosomes of postnatally stressed mice was found to be involved in glucose metabolism and fatty acid oxidation, which could facilitate altered glucose and insulin metabolism in the progenies of sires affected by similar postnatal stress treatment [76]. Together, the results from these studies suggested that, although the abundance of miRNAs remained fairly constant during the epididymal transition, the levels of individual miRNAs may be altered by epididymosomes in response to stress, which in turn modulate the miRNA profiles in the mature sperm.

Similarly, it has been postulated that epididymosomes mediate the alteration of tsRNAs/tRFs in response to ethanol exposure [107]. tRF-Glu-CTC was found to be significantly increased in both cauda epididymosomes and cauda sperm but not in caput sperm, suggesting that the tsRNAs/tRFs alteration in mature sperm stemmed from epididymal transit or storage through epididymosomes in the lumen. This was further confirmed by the level of tRF-Glu-CTC being elevated in the testicular spermatozoa after the co-incubation with cauda epididymosomes from ethanol-exposed mice. It is noteworthy that the phenotypes of the resultant offspring derived from IVF using naive oocyte and cauda sperm incubated with ethanol-exposed epididymosomes did not fully recapitulate the phenotypes of those from natural mating [108,109]. These results suggest an additional layer of regulation on the sperm sncRNA transcriptome in other parts of the male reproductive tract. Interestingly, although tsRNAs/tRFs and their modifications are the major mediators of the paternal inheritance of metabolic traits, the question as to whether the tsRNAs/tRFs carried by epididymosomes and their modification status are altered in obesity or HFD conditions remains elusive. Nonetheless, the involvement of epididymosomes in the alteration of the sperm miRNA and tsRNA/tRF profiles under stress and ethanol exposure conditions, respectively, has indicated the involvement of epididymosomes in paternal epigenetic inheritance [112,114].

Although miRNAs and tsRNAs/tRFs are two major sncRNA species in sperm, their level in sperm may have been overlooked due to the technical limitation of traditional small RNA sequencing technology. By improving the library preparation that has been blocked by RNA modifications in traditional sequencing platforms, it has been recently shown that the repertoire of RNA species in sperm has been underestimated, particularly for tsRNAs and rRNAs [115]. Therefore, a comprehensive characterization of sncRNA profiles and their RNA modifications status in matured sperm and epididymosomes is required to investigate the contributions of epididymosomes, and perhaps EVs from other parts of the male reproductive tract, to the establishment of the sperm sncRNA transcriptome. To this end, it should be recalled that the sncRNA repertoire in testicular sperm could be attributed to the endogenous expression by sperm per se before the transcriptional silencing or to it being conveyed to elongated spermatids or spermatozoa after the removal of cytoplasmic content. Importantly, a recent study has characterized the EVs in the testis. These testicular EVs were efficiently taken up by elongated spermatids and spermatozoa, and the sncRNA cargoes were also found in matured sperm [116]. These results suggest the potential contributions of testicular EVs to the inheritable sperm sncRNA transcriptome.

## 6. Implications for Assisted Reproductive Technology

Hitherto, direct evidence for the paternal epigenetic inheritance mediated by the sperm sncRNA transcriptome was mainly obtained from mouse models, where researchers recapitulate the paternal inheritance by the co-injection of either total or a subpopulation of sperm RNAs from sires exposed to environmental insults, or specific sncRNA candidates identified from the differential expression profiling of the sires, to fertilize an oocyte and generate F1 offspring. Studies reporting on paternal epigenetic inheritance in larger animals are limited because of the following reasons. First, the intracytoplasmic sperm injection protocol is more challenging in larger animals. Second, the study of inheritable traits in larger animals requires a long turnover time that may take over a decade in order to observe the phenotypes. Third, there is a technical impracticability of controlling the effects of the environment on the parental animals or offspring, particularly in humans, where confounding variables such as diet and lifestyle cannot be well controlled. Nonetheless, similar alterations in the sperm sncRNA transcriptome in response to similar environmental insults have been characterized in larger animals and humans. Thus, it is expected that paternal epigenetic inheritance is to be observed in these mammalian species. In this case, the establishment of the sperm sncRNA transcriptome in the different parts of the male reproductive tract via EVs could have broad implications in ART for preventing the undesirable inheritance.

ART is a spectrum of fertility treatments involving the manipulation of gametes in a laboratory and the transferring of a viable embryo to the uterus. The two widely adopted methods for overcoming male factor infertility are conventional IVF and intracytoplasmic sperm injection (ICSI). The ICSI procedure is gaining tremendous popularity, as it is also applicable to patients with severe male infertility such as a low sperm count (oligospermia) or absolute poor sperm motility (asthenospermia) in the ejaculate, or even sperm obtained from a surgical retrieval that is therefore incapable of fertilizing the oocyte through conventional IVF. The International Committee for Monitoring ART reported a delivery rate of 21.2% for ICSI in comparison to that of conventional IVF (22.4%) out of more than 600 k ART cycles in 2002, which is a decade since the first ICSI [117]. In the follow-up in the latest report of 2017 data, the ICSI procedure accounts for 70% of all ART cycles, with a comparable delivery rate against conventional IVFs [118].

Despite being the more common choice of practice in ART nowadays, the debate regarding the sperm of choice for the ICSI procedure with confirmed male infertility still remains unsolved. Given that the ICSI procedure requires only a 1:1 ratio of sperm and oocytes, patients are offered either testicular or epididymal sperm extraction for their ICSI cycle in the quest of selecting the ‘best sperm’ with optimal characteristics in order to maximize the chance of successful fertilization, pregnancy and the delivery of a healthy new life. The general consensus favors epididymal sperm in cases of obstructive azoospermia (OA) with normal spermatogenesis [119,120], as they would have gone through the maturation process in the epididymis, where they acquire fertilizing ability [121,122]. Although studies showed that epididymal sperm have a higher fertilization rate [123,124] and clinical pregnancy rate [125,126,127,128] with OA patients when compared to testicular sperm in ICSI, possibly owing to the mature status of said spermatozoa, the overall statistics showed no significant difference between the two choices of sperm in OA patients [129]. An older study presented higher live birth rates when using epididymal sperm retrieved through microsurgical epididymal sperm aspiration (MESA) compared to Testicular Epididymal Sperm Extraction (TESE) [130], whereas newer retrospective studies showed higher live birth rates using testicular sperm but ultimately no significant difference between the two sperm sources in OA [124,126,128] and non-obstructive azoospermia (NOA) [127].

While congenital malformation including the cryptorchid testes, inguinal hernia and hypospadias had been reported in newborns from ICSI with testicular spermatozoa [131,132], studies showed no significant increase in birth defects between the use of testicular sperm and epididymal sperm [127,128], as well as between the use of testicular sperm and/or epididymal sperm and ejaculate sperm in ICSI [132].

Otherwise, the retrospective study showed that the use of testicular sperm resulted in significantly higher clinical pregnancy and live birth rates compared to ejaculate semen in NOA with previous failed ICSI [133]. The pregnancy rate was also significantly higher using testicular sperm compared to ejaculate in patients with complete asthenospermia [134]. A meta-analysis of ICSI among those with cryptozoospermia also showed that testicular sperm led to a significantly higher embryo quality, implantation rate and clinical pregnancy rate compared to ICSI using ejaculate semen, although there was no significance in the fertilization rate [135]. These qualities can potentially be explained by the higher sperm DNA fragmentation rate (SDF) of ejaculate semen [133,136,137,138,139,140,141]. Overall, studies showed that both testicular and epididymal sperm can be used for ICSI with comparable clinical outcomes.

## 7. Future Directions

In the previous sections, we reviewed the evidence from the literature in which environmental insults lead to altered sperm sncRNA profiles and mediate the transgenerational inheritance of metabolic traits to the offspring. Notably, the tsRNAs/tRFs and miRNAs that mediate the paternal inheritance of metabolic traits and psychological traits in HFD and trauma conditions, respectively, are both being conveyed to sperm by epididymosomes during the epididymal maturation. If the alteration in the sperm sncRNA transcriptome was mainly attributed to the epididymosomes, the immature sperm in the testis or the initial segment of the epididymis should be void of these changes or a relatively naive sncRNA transcriptome. Since testicular/immature sperm are capable of fertilizing the oocyte in ART with a comparable pregnancy outcome, the use of testicular/immature sperm in ART represents a feasible approach to intervening or circumventing the paternal epigenetic inheritance. Since the semen or sperm quality, e.g., ROS levels and SDF rate, are sensitive to the environmental insults [142,143], and a modest alteration in the miRNA profiles of the testis in response to environmental insults was observed [90,99], whether testicular or immature sperm is the optimal cell type for ICSI in the conditions where bypassing the paternal epigenetic inheritance caused by environmental insults to the offspring is desirable requires further investigation.

We call for investigations into the effect of various environmental insults on the sncRNA transcriptome of sperm in the testis, epididymis and ejaculates in animal models and for clinical studies to investigate the phenotypic traits of the offspring conceived through ICSI with testicular and epididymal/ejaculate spermatozoa from cohorts of men with diverse health statuses. Prospective studies in ICSI including physical and mental health data, the sperm sRNA transcriptome and the subsequent neonatal health can become invaluable in furthering the current knowledge of epigenetic inheritance in the human cohort on top of animal models.

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
