# Peer review of "The Sperm Small RNA Transcriptome: Implications beyond Reproductive Disorder"

_ijms, 2022, doi:10.3390/ijms232415716_

Round 1

Reviewer 1 Report

Recently, non-coding RNAs have emerged as new players in sperm regulation, and they could have a significant role postfertilization. This excellent review summarizes current knowledge on the small noncoding RNAs (sncRNAs) carried by the sperm (sperm sncRNA transcriptome). Authors discuss how sncRNA change in different environmental situations and their effects on fertility and the offspring. Authors also summarizes and discuss their implication in assisted reproductive technologies. This is an excellent review, well oriented and written. To the best of my knowledge there isn’t any key reference without quotation. Consequently, my advice is to publish it as is.

Author Response

We would like to thank the reviewer for the constructive comments.

Reviewer 2 Report

In this study the Authors gave a review on the role of sperm small non-coding RNAs (sncRNAs) in reproductive functions, and the effects of environmental insults on sncRNA transcriptome. It should be emphasized that numerous studies have already reported about the role of sperm scnRNAs in reproductive dysfunctions and epigenetic inheritance, and this review does not contribute anything new about the above-mentioned subjects. Furthermore, it has been confirmed that sperm-borne sncRNAs are sensitive to environmental conditions, and could play a role in the inheritance of paternally acquired mental traits. The Reviewer suggests that the effects of differences in lifestyle and environment factors, such as nutrition (obesity), stress, and air pollution, on DNA methylation and scnRNA transcriptome have been extensively discussed in a wide variety of published papers, particularly in those based on the mouse models.  

Author Response

We would like to thank the reviewer for the constructive comments. Indeed, a number of excellent papers have reviewed the previous findings on sperm sncRNAs and their effect on fertility/epigenetic inheritance. Review papers that focus on the sperm epigenome and inheritance often include all epigenetic mechanisms (DNA methylation, histone and chromatin modifications, sncRNA), and often have brief/general reports of the key papers emphasizing on offspring phenotypes as a result of sperm sncRNA in the mouse studies.

One review paper (Klastrup 2018 doi.org/10.1007/s00438-018-1492-8) did thoroughly review a few key papers from the 2013-2016 period in the HFD category, while our paper included more recent studies that emphasized the differences in establishing the mouse models, the common sncRNA candidates observed between the studies and their functionalities that could potentially explain the observed phenotypes in the offspring, which is not often included in published papers on paternal epigenetic inheritance or sperm small RNA papers. Another novelty of our review paper is that we relate the common targets observed in the sperm of mice to sperm data of men according to the environmental insults studied.

Lastly, other review papers call to elucidate the functional roles and mechanisms behind the sperm epigenome/sncRNAs as a means to understand epigenetic inheritance as well as potential biomarkers, which certainly are important. In this regard, our review includes discussions on the establishment of sperm small RNA transcriptome, which provide insight into epigenetic inheritance. Stemming from the discussion on the origin of sperm small RNA transcriptome, our review also provokes the idea of studying the testicular sperm as a method to circumvent the inheritance of undesirable traits stemming from environmental insults during the paternal lifetime in the event where ART is used to overcome male-factor infertility.

Once again, we are fully aware of the many published reviews discussing this important observation. We believe that while it is necessary to report the current knowledge obtained from animal studies, it is also important to relate them back to the human cohort. In our case, we discuss the knowledge with the intention of relating the paternal epigenetic inheritance towards ART and call attention to sperm sources in order to circumvent currently known inheritable traits stemming from environmental insults from animal studies. Therefore, we believe the review would bring a new perspective to the field.

Reviewer 3 Report

This review aims to summarize the current knowledge of the sperm sncRNA transcriptome, its alterations in response to environmental insults, and potential implications to ART. The subject matter is very topical and interesting, and it is crucial to understand and summarize current knowledge from this area of research. The authors appropriately combined information from older and recent references.

Minor:

Line 36: This review „first“ describes…. I found more reviews with similar interests (e.g. Luo, J. IJMS 2022; Yang, Andrology, 2022). Please, don’t use “first”

Lines 150-152: no Reference

Lines 204-214: no Reference

Author Response

We would like to thank the reviewer for the constructive comments. The minor concerns listed below have been revised accordingly.

Line 36: This review „first“ describes…. I found more reviews with similar interests (e.g. Luo, J. IJMS 2022Yang, Andrology, 2022). Please, don’t use “first”

Response: The text has been revised as suggested.

Lines 150-152: no Reference

Lines 204-214: no Reference

Response: The suggested references have been added.

Round 2

Reviewer 2 Report

The Reviewer still thinks that this paper does not offer any new information about the discussed subject matter, and it would been more helpful if the Authors had included the results of their previous findings.